# Hookworms dynamically respond to loss of Type 2 immune pressure

Annabel A. Ferguson[1], Juan M. Inclan-Rico[1], Dihong Lu[2], Sarah D. Bobardt[3], LiYin Hung[1], Quentin Gouil[4,5], Louise Baker[4,6], Matthew E. Ritchie[4,5], Aaron R. Jex[4,6], Erich M. Schwarz[6,7], Heather L. Rossi[1], Meera G. Nair[3], Adler R. Dillman[2], De'Broski R. Herbert[1]*

1 University of Pennsylvania, School of Veterinary Medicine, Pathobiology Department, Philadelphia, Pennsylvania, United States of America, 2 University of California Riverside, Department of Nematology, Riverside, California, United States of America, 3 University of California Riverside, School of Medicine, Department of Biomedical Sciences, Riverside, California, United States of America, 4 The Walter and Eliza Hall Institute of Medical Research, Parkville, Victoria, Australia, 5 University of Melbourne, Department of Medical Biology, Parkville, Victoria, Australia, 6 University of Melbourne, Department of Veterinary Biosciences, Parkville, Victoria, Australia, 7 Cornell University, Department of Molecular Biology and Genetics, Ithaca, New York, United States of America

* debroski@vet.upenn.edu

**Data Availability Statement:** The N. brasiliensis genome assembly and its sequencing data have been deposited in GenBank under the BioProject accession number PRJNA994163, with the genome assembly deposited as

## Abstract

The impact of the host immune environment on parasite transcription and fitness is currently unknown. It is widely held that hookworm infections have an immunomodulatory impact on the host, but whether the converse is true remains unclear. Immunity against adult-stage hookworms is largely mediated by Type 2 immune responses driven by the transcription factor Signal Transducer and Activator of Transcription 6 (STAT6). This study investigated whether serial passage of the rodent hookworm *Nippostrongylus brasiliensis* in STAT6-deficient mice (STAT6 KO) caused changes in parasites over time. After adaptation to STAT6 KO hosts, *N. brasiliensis* increased their reproductive output, feeding capacity, energy content, and body size. Using an improved *N. brasiliensis* genome, we found that these physiological changes corresponded with a dramatic shift in the transcriptional landscape, including increased expression of gene pathways associated with egg production, but a decrease in genes encoding neuropeptides, proteases, SCP/TAPS proteins, and transthyretin-like proteins; the latter three categories have been repeatedly observed in hookworm excreted/secreted proteins (ESPs) implicated in immunosuppression. Although transcriptional changes started to appear in the first generation of passage in STAT6 KO hosts for both immature and mature adult stages, downregulation of the genes putatively involved in immunosuppression was only observed after multiple generations in this immunodeficient environment. When STAT6 KO-adapted *N. brasiliensis* were reintroduced to a naive WT host after up to 26 generations, this progressive change in host-adaptation corresponded to increased production of inflammatory cytokines by the WT host. Surprisingly, however, this single exposure of STAT6 KO-adapted *N. brasiliensis* to WT hosts resulted in worms that were morphologically and transcriptionally indistinguishable from WT-adapted parasites. This work uncovers remarkable plasticity in the ability of hookworms to adapt to their hosts, which may present a general feature of parasitic nematodes.

GCA_030553155.1, and with RNA-seq reads being deposited in the NCBI Sequence Read Archive (SRA) under the BioSample accession numbers SAMN37187590, SAMN37187592, SAMN37187594, SAMN37187619, SAMN37187621, SAMN37187623, SAMN37203688, SAMN37203689, SAMN37203690, and SAMN37203691. The exact protein-coding gene predictions, coding DNAs (CDS DNAs), and predicted protein sequences used in this analysis prior to GenBank submission have been permanently archived in OSF at https://osf.io/gpyuc.

**Funding:** Computing was enabled by a research allocation from NSF XSEDE (TG-MCB190010) to EMS. AAF was supported by an NIH T32 (AI007532-25). Funding for this work was awarded to EMS (Cornell startup funds), MGN and ARD (NIH R21AI142121), and to DRH (NIH R01AI164715 U01AI163062). ARJ is supported by the Australian National Health and Medical Research Council (Investigator Grant APP1194330). The funders had no role in study design, data collection and analysis, decision to publish, or preparation of the manuscript.

## Author summary

Hookworms infect 170–440 million people globally. Our goal in this study was to understand how the host immune environment affects hookworm fitness and gene expression. It is well accepted that protective immunity against hookworms depends on mechanisms controlled by the Signal Transducer and Activator of Transcription 6 (STAT6). Therefore, we used mice lacking STAT6 (STAT6 KO) to serve as a permissive environment for survival of the rodent hookworm *Nippostrongylus brasiliensis*. One passage into STAT6 KO hosts resulted in immediate transcriptional changes, including increased feeding and egg production pathways, corresponding with measured increases in fecundity, survival, body size, and feeding observed after additional passages. It took multiple generations (up to 15) of STAT6 KO hosts to see the reduced transcription of genes thought to mediate evasion of the host immune system. Despite WT hosts having an enhanced immune response to STAT6 KO-adapted *N. brasiliensis*, this line of worms survived as well as the WT-adapted *N. brasiliensis* and quickly lost the changes they had acquired during passage through STAT6 KO hosts. Our findings suggest that hookworms dynamically alter their transcriptional profile to suit their immediate host immune environment.

## Introduction

Successful parasites often co-evolve with their hosts over time through a process that achieves an optimal balance between pathogen virulence and host survival. In this way, host immune responses can be a key driver of coevolution. Numerous studies have described adaptive mechanisms for parasite evasion and manipulation of host immunity, with ensuing fitness benefits to the pathogens [1]. For eukaryotic metazoans like parasitic helminths, numerous effector mechanisms have been described to have direct immunomodulatory functions [2–6]. However, few studies have focused on how host immunity impacts the evolution of helminth parasite fitness due to the multicellular nature of helminths and complex helminth life-cycles that often require multiple hosts for successful experimentation [3]. Although progress has been made in the genetic manipulation of certain helminth species, the molecular and biochemical characteristics of host adaptation remain poorly understood [7–9].

Experimental evolution, which can identify adaptive traits, is a controlled study of processes that drive species evolution within a setting wherein environmental parameters and the subject population are tightly controlled [10]. This approach is borne out of a quintessential study from Richard Lenski's group showing that multiple diverse and convergent genetic traits arise and go extinct in *E. coli* upon adaptation to differing media conditions [11,12]. Accordingly, pathogen adaptation to host immunity is often studied by serial passage within a well-defined host or set of culture conditions. Serial passage experiments centered on the gastrointestinal (GI) parasitic nematode *Heligmosomoides polygyrus* revealed improved survival and reproductive output within immunized wild-type mice following six or more generations of selection [13]. Moreover, selecting late- versus early-infection eggs of the rat-adapted GI parasite *Strongyloides ratti* resulted in improved parasite fitness and transcriptional changes of late-stage worms inoculated within immunized WT hosts [14,15]. However, it remains untested whether GI parasites, particularly hookworms, undergo phenotypic or transcriptional adaptations upon parasitism of an immunocompromised host, particularly a host deficient in the type of immunity that may have specifically evolved to eliminate multicellular parasitic helminths [16]. A recent case study suggests that host specificity boundaries can expand if parasites encounter immunosuppressed hosts [17].

*Nippostrongylus brasiliensis* is a rat hookworm that exhibits similarities in life cycle and blood-feeding to human hookworms [18,19]. *N. brasiliensis* infects hosts by penetrating the skin, migrating through the vasculature, migrating through the lungs at day 2 post-infection, presumably coughed-up and swallowed for entry into the GI tract by day 3, affixing itself to the GI tract, and becoming sexually mature diecious adults at day 5 that reproduce and produce eggs until being expelled by the host [20]. The key mechanism for host protection against hookworms in humans and mice is Type 2 immunity, characterized by CD4$^+$ T helper cells, innate lymphoid group 2 cells, eosinophils, mast cells, IgE, and the interleukins IL-4, IL-5, IL-9, IL-13, IL-25, and IL-33. Collectively these responses target different stages of worm infection and limit their survival through many mechanisms. Studies of human hookworm infection have shown that IL-5 production correlates with protection from re-infection following anthelminthic treatment [21], and that experimental or natural infection is characterized by eosinophilia and the production of IgE and IgG antibodies specific for hookworm antigen [22]. Despite this, immunity is not sterilizing or lasting, and people remain susceptible to reinfection following anthelmintic treatment [23–25]. In contrast to other helminth infections, hookworm burden in people increases from early childhood to adulthood, and remains high in adults [26]. This suggests that hookworms may uniquely possess the ability to respond adaptively to hosts with differing levels of immune pressure.

Mouse studies show that IL-4/IL-13 receptor signaling is crucial for hookworm clearance and that this receptor recruits the canonical transcription factor STAT6 [27]. STAT6 binds the promoter region of many Type 2 immunity target genes necessary for effector responses that eliminate the worms in the lung and intestine [28,29]. In the *N. brasiliensis* model, mice deficient in STAT6 fail to clear worms by day 11 to 15, likely due to the known importance of STAT6 in driving IL-4/IL-13 signaling downstream of IL-4Rα, resulting in mucus accumulation, M2 macrophage differentiation, smooth muscle contraction, and tuft cell expansion [27,30–38]. However, it has been unclear whether STAT6 deficiency in the parasitized host affects hookworm phenotypes like size, feeding capacity, or gene expression in hookworms.

This study tested whether the experimental evolution of *N. brasiliensis* by sequential passage through STAT6 KO hosts would lead to a functional, potentially heritable impact on overall worm fitness when reintroduced back into WT hosts. Results show a dramatic increase in worm size, fecundity, and blood feeding upon adapting hookworms to STAT6 KO mice within passage hosts. These changes occurred at the apparent expense of expressing transcripts encoding diverse signaling and developmental pathways. Interestingly, after 15 generations, we noted a marked decline in transcripts encoding proteases, SCP/TAPS proteins, and transthyretin-like proteins; these categories of proteins are frequently observed in excreted/secreted proteins (ESPs) of hookworms and other parasitic nematodes that are suspected of having an immunomodulatory function [39,40]. Indeed, when inoculated into WT hosts, STAT6 KO-adapted parasites elicited increased Type 1, Type 2, and Type 3 inflammatory responses. However, there was little evidence that any heritable adaptations occurred within STAT6 KO-adapted worms because, after being inoculated into WT hosts, they survived as well as WT-adapted worms and reverted their transcriptional profile to that of WT-adapted worms. These results imply that releasing the selective pressure of Type 2 immunity may alter some of the host-modulatory traits of hookworms, but that hookworms rapidly adjust their phenotype and gene expression profile to suit the immediate host immune response.

# Results

## *N. brasiliensis* develops increased reproductive fitness in STAT6 deficient mice

STAT6-dependent immune activities are required to clear *N. brasiliensis* from rodent hosts [27]. We hypothesized that the STAT6 KO immune environment would increase the

reproductive fitness of *N. brasiliensis* relative to the WT immune environment. To address this question, we isolated 500 infectious stage larvae (iL3) from WT rat-maintained *N. brasiliensis* (parental *N. brasiliensis*), infected WT or STAT6 KO mice with them, and then serially passaged subsequent iL3 derived from those mice to generate WT- or STAT6 KO-adapted *N. brasiliensis* (Fig 1A). We found that following 16 or 15 generations of selection in WT or STAT6 KO, respectively, STAT6 KO-adapted *N. brasiliensis* had increased eggs per gram of feces (EPG), with a greater than 10-fold increase of area under the curve (AUC; i.e., total observed EPGs) (Fig 1B and 1C). Because EPG could be determined by either the number of reproductive-stage worms in the small intestine or the fecundity of the individual female worms, we sought to determine whether STAT6-driven immunity had an impact on individual worm fecundity. Individual female worms were isolated and placed into culture media, and eggs released per individual were counted after 24 hours. We found an increase in 24-hour egg release of worms at day 7 post-infection in STAT6 KO-adapted worms relative to WT control worms (Fig 1D). In addition, the total number of adapted adult worms within STAT6 KO hosts increased relative to WT controls at days 7 and 8 (Fig 1E and 1F). This implies that STAT6 KO-adapted hookworms have increased survival and fecundity within STAT6 KO hosts.

## Adult stage STAT6 KO-adapted hookworms have increased size, ATP, and hemoglobin content relative to WT-adapted parasites

IL-4Rα-dependent STAT6 signaling is necessary for goblet-cell secretion of RELM-β, which interferes with the feeding capacity and health of GI nematodes [31]. Experiments were performed to understand whether STAT6 KO-adapted worms developed abnormal morphological or biochemical features. Data in Fig 2A show that STAT6 KO-adapted females qualitatively appeared more red in color and larger than WT-adapted females. Measurement of individually isolated fecund adult females at day 7 post-infection revealed that the STAT6-deficient environment also resulted in significantly longer worms (Fig 2B). Hemoglobin levels were then measured within worm homogenates to assess whether the red color and increased size corresponded with a greater feeding capacity. Hookworm hemoglobin levels in STAT6 KO-adapted worms were significantly higher than in WT controls (Fig 2C). This difference was lost when measurements were adjusted to average worm length (Fig 2D), indicating that nutritional intake was proportional to body size. The increased nutritional intake of STAT6 KO-adapted worms suggests that they would also have increased available energy. We therefore measured ATP within worm homogenates (10 worms per measurement) and found significantly increased ATP concentration in STAT6 KO-adapted and derived worms relative to WT controls at day 7 post-infection (Fig 2E). Worm ATP was also proportional to the size, given that the magnitude increase was similarly diminished after normalizing for worm length (Fig 2F). Collectively, these results suggest that STAT6-driven immune responses impede the size and feeding capacity of hookworms.

## WT- and STAT6 KO-adapted worms have distinct gene expression profiles within their passage hosts

Based on the dramatic increase in *N. brasiliensis* size, survival, and reproductive fitness when the worms reside in STAT6 KO hosts, we hypothesized that these parasites may also alter their transcriptional profiles in this immunodeficient environment. This alteration could occur immediately or require several generations of passage. To find out whether worms exhibit an altered transcriptional response upon a single generation of exposure to STAT6 KO hosts, we performed single-worm mRNA-seq on progeny from parental (WT rat-adapted) *N.*

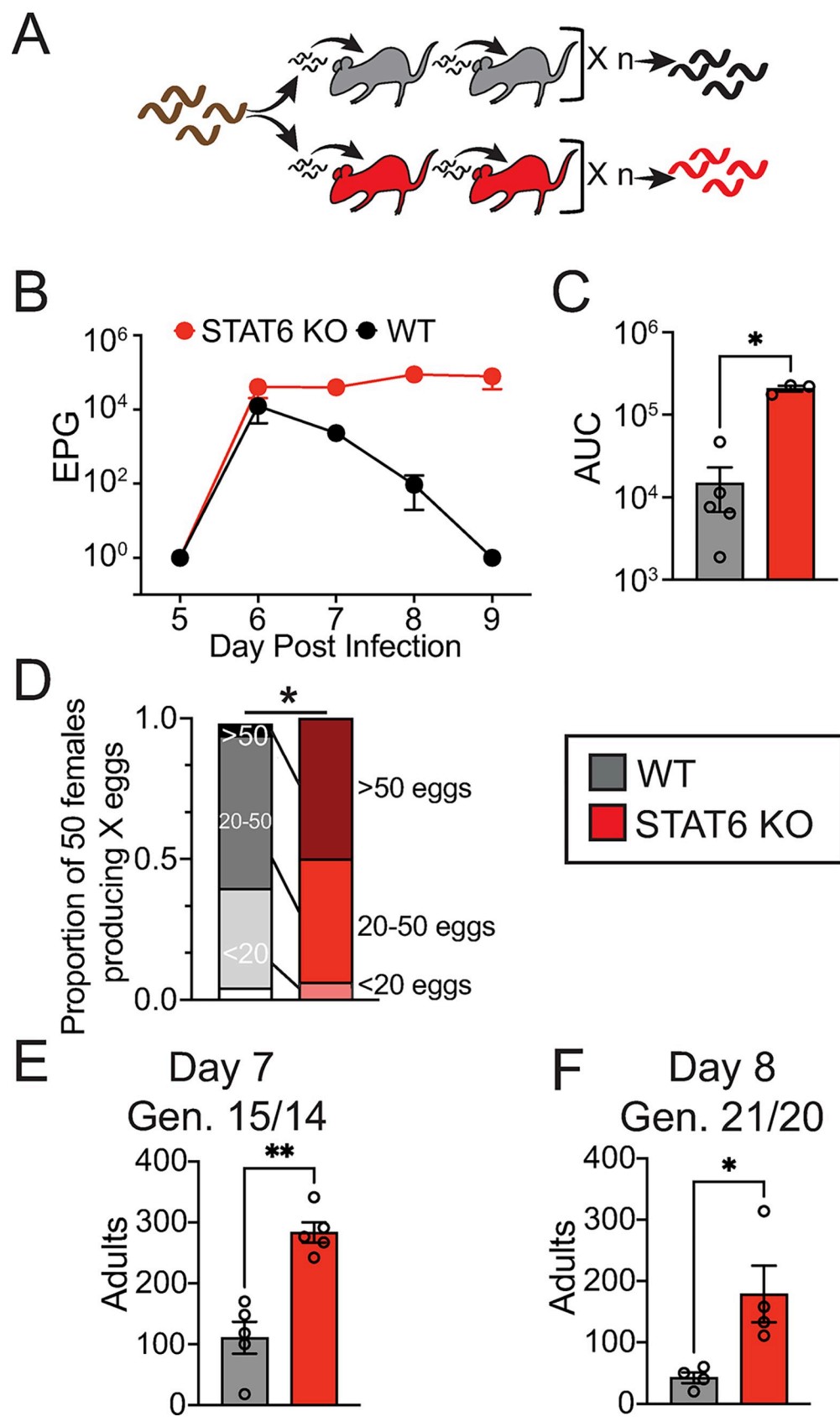

**Fig 1. STAT6-driven immunity decreases *N. brasiliensis* reproductive output and survival within host. A.**
Experimental evolution design, *N. brasiliensis* (originally isolated from rat host, brown) were infected for several generations (n) through either WT (black or gray) or STAT6 KO (red) mouse hosts to produce adapted cohorts. Data in this figure is from either the 11th (D, individual female egg output in 24 hours), 15th (E, intestinal adult worms), 16th (B,C–eggs per gram of feces, EPG, output), or 21st (F, intestinal adult worms) infection generation of WT *N. brasiliensis* and the 10th, 14th, 15th, or 20th generation of STAT6 KO *N. brasiliensis*. **B.** EPG from each line were assessed for days 5–9 within their passage host genotype. **C.** Area under the curve (AUC) of EPG from days 5–9 by host genotype. **D.** 50 adult female *N. brasiliensis* from each line were isolated from the intestines of their passage host and assessed for 24-hour egg-release *ex vivo*. Shown are the proportions of 50 adult worms per condition releasing 0, <20, 20–50, or >50 eggs. Note that *N. brasiliensis* females from STAT6 KO all produced some level of eggs and significantly more of them produced more than one egg each, as compared to WT. **E.** Number of adults per host isolated from the small intestine at day 7 post infection within passage-hosts (generations 15 or 14). **F.** Number of adults per host isolated from the small intestine at day 8 post infection within passage-hosts (generations 21 or 20). These are the results of two replicate experiments with 3–5 mice/genotype each. Unless otherwise indicated, data are individual values (circles) with mean ± SEM indicated (bars), * p < 0.05, **p < 0.01, by t-test (C, E, F) or Chi-squared test (D).

*brasiliensis* infections of either WT or STAT6 KO mice isolated at day 2 from the lung (L4), as well as days 5 (D5 adult female) and 7 from the gut (D7 adult females) (Fig 3A). We also performed single-worm mRNA-seq on the two adapted lines, derived from passage hosts at G15 (WT) and G14 (STAT6 KO), isolating adult females at day 7 post-infection (Fig 3B). In the G1 infections, individuals were first distinguishable by life-cycle stage and host genotype using PCA and hierarchical clustering (Fig 3C and 3E). Day 5 (D5) and Day 7 (D7) adult females from STAT6 KO hosts clustered independently from the equivalent stage in WT hosts, which was not evident for the L4 stage by PCA plot (Fig 3C). Hierarchical clustering also indicated that day 7 adults could also be separated by host genotype (Fig 3E). Likewise, in the adapted worm lines, WT (G15) or STAT6 KO (G14) groups each produced distinct transcriptional profiles as indicated by both PCA and hierarchical clustering (Fig 3D and 3F). For analyzing all RNA-seq data, we generated an improved genome assembly and protein-coding gene set for *N. brasiliensis* with higher completeness (proteome BUSCO score of 97.5%) than in previously published work (proteome BUSCO score of 75.1%) [41].

In accordance with clustering patterns, only 14 out of 16,464 protein-coding genes were differentially expressed in the L4 stage between WT and STAT6 KO hosts at G1 (Fig 3G). By day 5 post-G1 infection, slightly more (106) differentially expressed genes were identified in D5 female worms from the gut (Fig 3H). No differences in L4 stage parasite burden in the lung (24 or 48 hours post infection), or day 5 adult worms were observed between WT and STAT6 KO hosts (S1 Fig). Among day 7 adult female worms, far more genes (2,909) were differentially expressed, with 97% of these genes (2,815) being downregulated in the STAT6 KO condition (Fig 3I). Interestingly, the introduction of parental stock worms into STAT6 KO mice (G1) resulted in 12.7 times as many differentially downregulated genes compared to G15/14 host-adapted worms (2,815 genes downregulated in G1 versus 222 genes downregulated in G14 STAT6 KO; Fig 3I and 3J). Surprisingly, among the significantly up- or downregulated genes (log2FoldChange > 2 or < - 2 and padj < 0.05), the numbers of shared genes were limited, and indistinguishable from overlaps that would have been expected by chance (S2A Fig). In addition, while most genes show concordant expression in the STAT6 KO condition between G1 or G15 adapted worms (1,004 genes), a substantial proportion were discordant between the two datasets (330 genes; S2B Fig). This suggests that the hookworm transcriptional response to mouse STAT6-driven immunity is more dynamically altered without prior generational exposure to such hosts. Moreover, hookworms had distinct transcriptional profiles in hosts lacking STAT6-driven immune responses, which were apparent as early as day 5 post-infection, and by day 7 show a bias towards overall suppression of gene expression in STAT6 KO hosts.

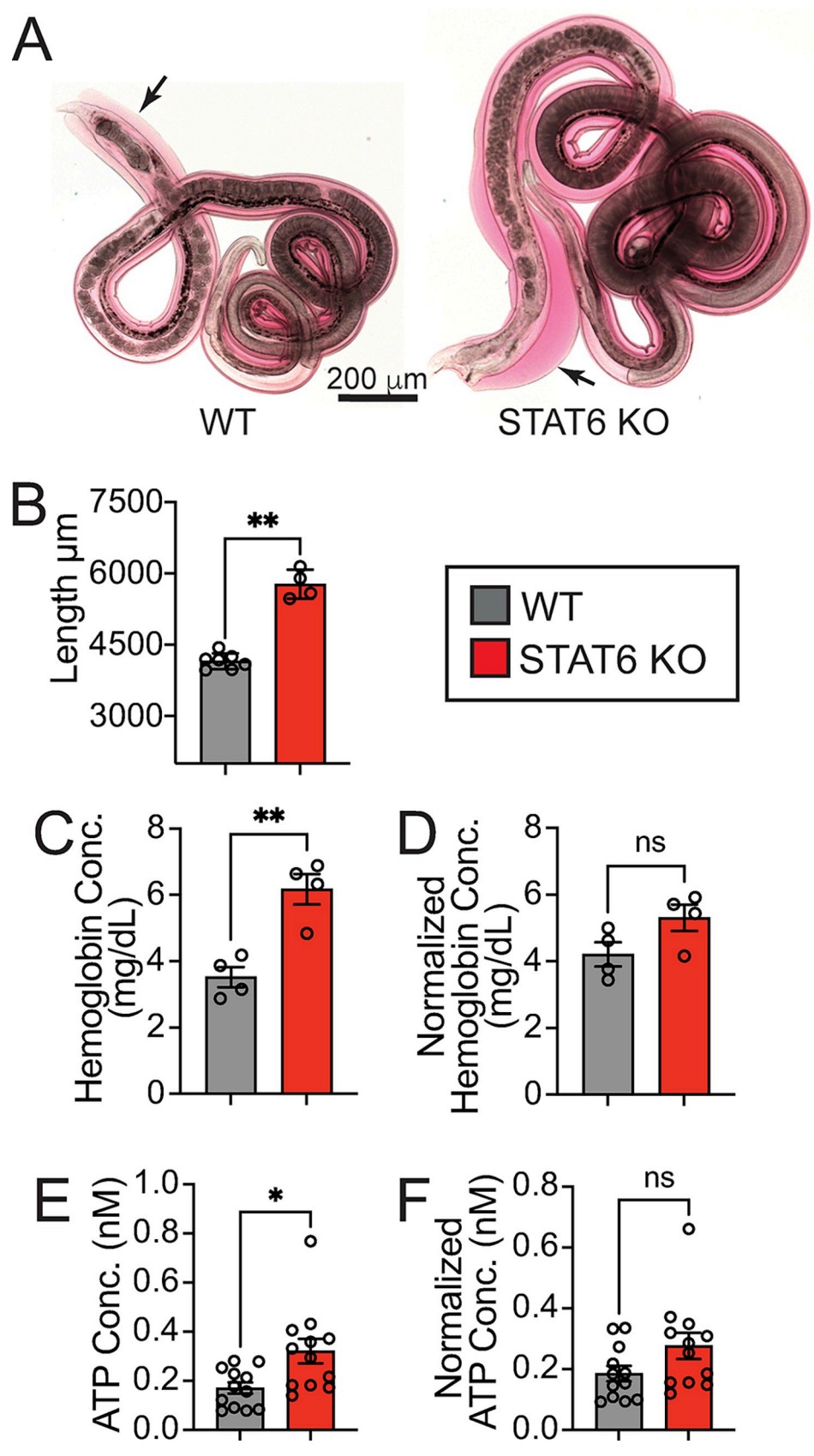

**Fig 2. *N. brasiliensis* exhibits increased size, hemoglobin, and ATP content, in the absence of STAT6-driven immune responses. A.** 10x magnification images of adult female WT or STAT6 KO-adapted worms (generation 23 and 22) derived from passage hosts at day 7 post-infection (scale bars = 200 μm, arrows indicate edge of worms). Data in B-F are from adult worms isolated at day 8 post-infection from the 21st (WT) or 20th (STAT6 KO) generation. **B.** Female worm lengths of WT- or STAT6 KO-adapted worms derived from passage hosts. **C-D.** Hemoglobin concentration or worm length-normalized concentration of worms derived from passage hosts. **E-F.** ATP concentration or worm length-normalized concentration of worms derived from passage hosts. Unless otherwise indicated, data are individual values (circles) with mean ± SEM indicated (bars), * p < 0.05 ** p < 0.01, *** p < 0.001, by t-test.

## Adult-stage hookworms express distinct gene families in the absence of STAT6-driven immunity

To further explore the functional role of genes that were differentially expressed in worms derived from WT versus STAT6 KO hosts, we performed gene set enrichment analysis (GSEA) [42,43] using multiple databases of gene set terms, including Gene Ontology (GO), biological function, cell compartment, and molecular function terms, InterPro protein domain names, and eggNOG description terms [44–46]. Many gene set terms of interest were identified as positively or negatively enriched in the STAT6 KO condition (FDR-adjusted p-value < 0.05) from G1 infections (Fig 4A and S1 Raw Data) and infections with the adapted (G15/14) cohorts (Fig 4B and S2 Raw Data). The most prominent upregulated gene sets in either G1 or adapted cohort infections were those associated with embryogenesis and cell division. Both G1 worms and G15/14 adapted worms from STAT6 KO infections had enriched expression of gene sets annotated with GO terms, including Cell cycle (GO:0007049), Gene silencing (GO:0016458), Meiotic cell cycle (GO:0051312), and Chromosome (GO:0005694); all of these biological annotations are associated with embryogenesis in nematodes [47]. Enrichment of these gene sets is consistent with our data showing that *N. brasiliensis* becomes more fecund when adapted to and derived from STAT6 KO hosts (Fig 4B).

The STAT6 KO-downregulated gene sets were interesting because we hypothesized that many of these represent functions involved in countering or evading a Type 2 immune response. The expression of genes putatively related to immune evasion might be suppressed in worms grown in a STAT6 KO environment in favor of increased embryogenesis. Indeed, diverse functional gene sets were down-regulated within the G1- (Fig 4A and S1 Raw Data) or G14-adapted *N. brasiliensis* worms (Fig 4B and S2 Raw Data) derived from STAT6 KO hosts relative to WT controls. Many were functionally similar in the G1 or adapted *N. brasiliensis* infections. Most striking were pathways that suggested a downregulation of specific behavioral responses and peptidase products. FMRFamide-related peptide-like [IPR002544] or Neuropeptide signaling (GO:0007218) were strongly down-regulated in both the single-generation and the adapted *N. brasiliensis* from STAT6 KO hosts (Figs 4, S3B, and S3C). Both gene sets are comprised of genes encoding FMRFamide-like peptides (FLPs), which are highly conserved among nematodes, show dynamic expression in both free-living and parasitic nematodes [48] and have been shown to control foraging, mate-seeking behaviors, and avoidance of noxious stimuli [49,50]. Downregulation of these genes is consistent with a reduced need for directed mobility within a host lacking the weep and sweep response responsible for dislodging worms from the mucosal tissue [31,32,51,52]. Other gene set families enriched in the WT condition suggested an increased response to stress, such as cytochrome p450 in the G1 infection and HSP20-like chaperone in the G15/14 infection. Lastly, gene categories most enriched in the WT condition of G15/14 infections included those encoding proteases (papain family cysteine protease, astacin, metallopeptidase, aspartic peptidase domain superfamily), SCP/TAPS proteins, and transthyretin-like proteins (Fig 4A and 4B). Proteases, SCP/TAPS, and

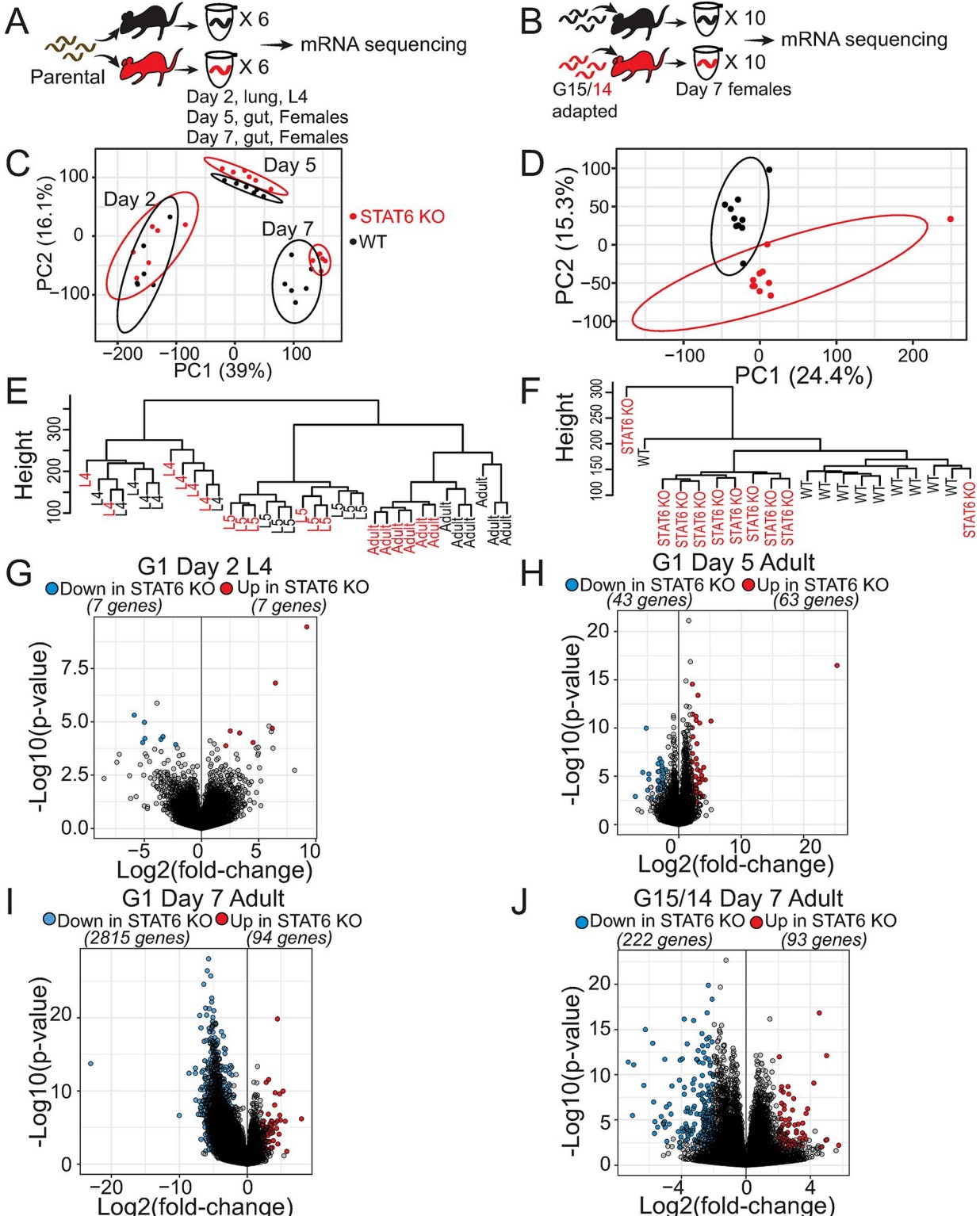

**Fig 3. *N. brasiliensis* has an altered transcriptional profile within STAT6 KO hosts. A.** Experimental design. Parental *N. brasiliensis* (brown) were used to infect WT (black) or STAT6 KO (red) mice, G1. mRNA was sequenced from individual worms isolated from either lung at day 2 post infection (L4), or female worms from the small intestine at day 5 (D5 Adult) or day 7 (D7 Adult) post infection (6 worms/stage and host genotype). **B.** WT (black) or STAT6 KO (red) adapted worms (generation 15 and 14, respectively) were isolated from passage hosts at day 7 post infection (10 adult female worms/host genotype). **C.** Principal component analysis (PCA) plot principal components (PC) 1 and PC2 for G1

individual worm by life cycle stage and host genotype. Each point represents a single-worm replicate. **D.** PCA plot of PC1 and PC2, for G15/14 individual adult female worm gene expression with ellipses of the 95% normal probability for each condition. Three outlier points were removed from the analysis (two from STAT6 KO condition and one from WT). **E-F.** Hierarchical clustering of gene expression data for G1 and G15/14 respectively. **G-I.** Volcano plots of differential gene expression results from worms derived from G1 *N. brasiliensis* infection of WT or STAT6 KO mice at L4, D5 or D7 adult female stage. **J.** Volcano plot of differential gene expression results of G15/14 adult female WT or STAT6 KO mouse adapted *N. brasiliensis* from passage hosts; significance indicated by Log2FC > 2 or < -2, and adjusted p < 0.05.

transthyretin-like proteins have been repeatedly observed among ESPs of parasitic nematodes, and such ESPs are either generally suspected or, in particular cases, known either to modulate the host immune response or to be involved in migration through host tissue [2,39,40,53–55]. When we directly analyzed the relative expression of these ESP-like genes, we found that they were significantly down-regulated in the STAT6 KO condition infected with adapted (G14) worms (Fig 5A and 5B). However, to our surprise, in infections with parental stock, we found a slight upregulation of such genes in the D7 adult stage and a minimal trend in the D5 adult stage (Fig 5C–5F). In addition, we isolated supernatant from adult worm cultures of WT- or STAT6 KO-adapted worms, which likely contained excreted/secreted products (both ESPs and other non-protein molecules), and tested their ability to elicit a cytokine response from mesenteric lymph nodes of hookworm infected WT mice. We found that the STAT6 KO-adapted worm derived supernatant resulted in decreased production of IL-4 but not IL-10 (S5 Fig). This suggests that STAT6 KO-adapted worms produce excreted/secreted products that are less immunogenic, perhaps having reduced antigenic diversity.

## STAT6 KO-adapted worms elicit a stronger inflammatory immune response from WT hosts

Given that gene categories associated with ESPs were downregulated in the STAT6 KO-adapted worms, perhaps corresponding to less secretion of immunomodulatory proteins, we hypothesized that WT hosts might have more robust immunity against STAT6 KO-adapted worms. To test this, we infected WT mice with STAT6 KO- or WT-adapted worms and assayed the cellular content of gut-draining mesenteric lymph nodes to assess adaptive immune responses or peritoneal lavage fluid to assess locally induced innate responses by flow cytometry (Figs 6A and S4). Lymph node-isolated cells were stimulated with PMA and assayed for cytokine expression by flow cytometry. Mesenteric lymph node (MLN) cells were gated to identify lymphoid cells like T helper cells, γδ T cells, and CD8 cytotoxic T cells (Figs 6B and S4). Strikingly, we found a dramatic shift in immune cell proportion and number in WT mice infected with STAT6 KO-adapted worms. The total number of MLN cells was significantly increased in the STAT6 KO infection condition (Fig 6C), suggesting a stronger elicited immune response. This response was characterized by an increased proportion and number of IL-13+ and IL-33 receptor ST2+ positive Th2 cells within the mesenteric lymph nodes (Fig 6D–6G). In addition, STAT6 KO-adapted worms elicited a significant increase in both percentage and number of IL-17+ γδT and IFNγ+ CD8+ T cells (Fig 6H–6K). Accordingly, MLN isolated from these animals secreted more IFN-γ than WT-adapted worms when stimulated with anti-CD3/CD28 (S5 Fig).

To assess the local myeloid lineage effector cells likely elicited by cytokine production, we looked in the lavage fluid of the peritoneal cavity (Fig 6L). We found an increase in FcεR positive cells in the STAT6 KO *N. brasiliensis* infection condition, indicating more cells that may bind to IgE antibodies (Fig 6M and 6N). We also found an overrepresentation of the total macrophage and the Type 2 associated macrophage population in the STAT6 KO-adapted *N. brasiliensis* infection condition, the latter of which expresses Arginase 1 (Arg1) a known M2 macrophage marker population (Fig 6O–6R). However, we did not see an increase in

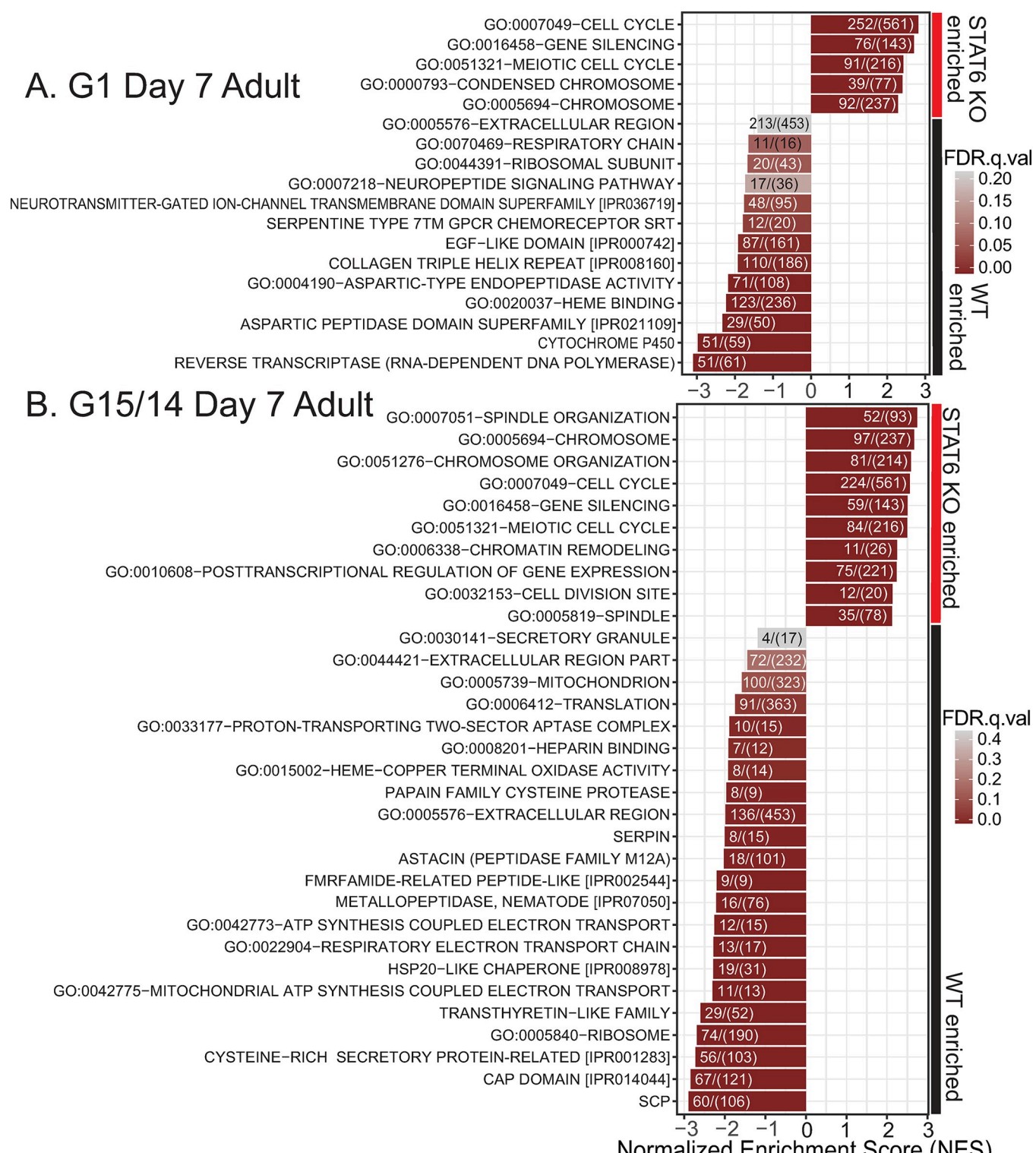

**Fig 4. Host STAT6 immunity affects *N. brasiliensis* gene expression of common families. A-B.** Selected top up- and downregulated gene family terms identified in gene set enrichment analysis (GSEA), of WT- or STAT6 KO-derived adult day 7 female worms, from an infection with (**A**) parental *N. brasiliensis* stock, G1, or (**B**) WT (generation 15) or STAT6 KO (generation 14) adapted *N. brasiliensis* stock.

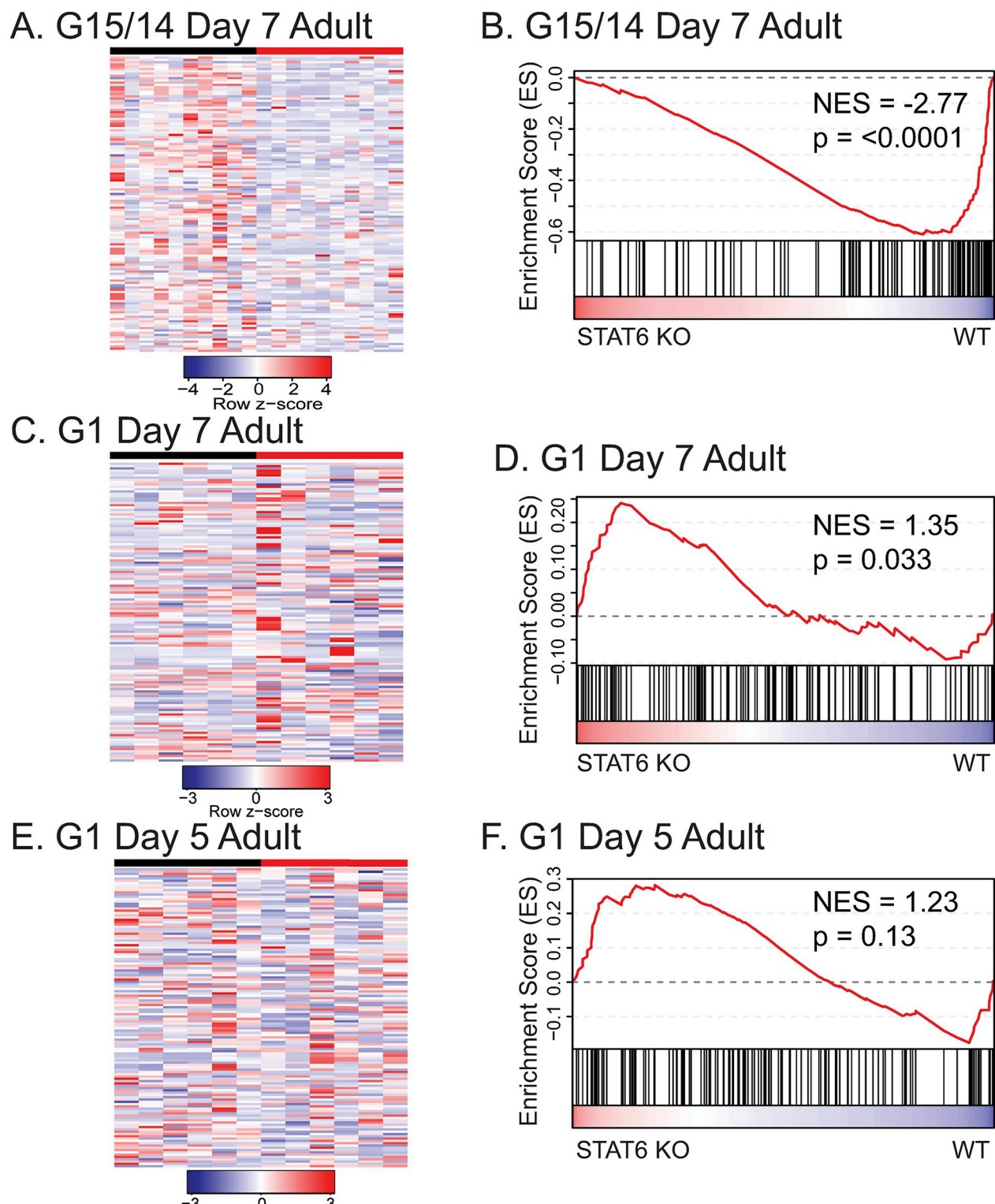

**Fig 5. G14 STAT6 KO-adapted *N. brasiliensis* down-regulate gene categories associated with ESPs. A-B.** Heatmap and GSEA enrichment plot of transcripts encoding gene categories associated with ESPs for WT or STAT6 KO generation 15/14 adapted female *N. brasiliensis* derived from passage host at day 7 post infection. **C-F.** Heatmap or GSEA enrichment plot of transcripts encoding adult stage ESP-associated gene categories for G1 *N. brasiliensis* infection of WT or STAT6 KO mice, **(C-D)** of day 5 post infection female worms or **(E-F)** day 7 post infection female worms.

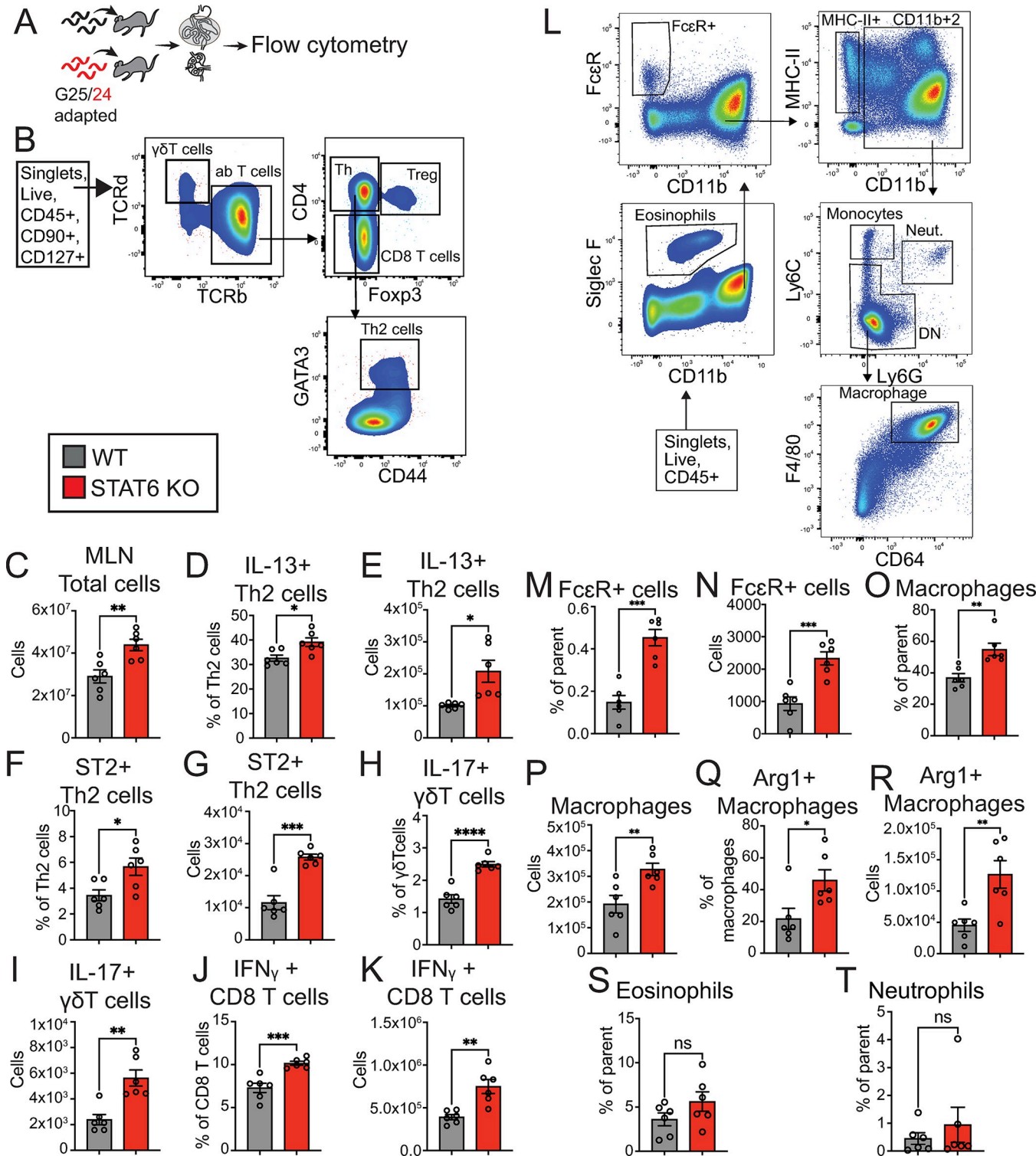

**Fig 6. STAT6 KO-adapted *N. brasiliensis* elicit a stronger inflammatory WT host immune response.** Mesenteric lymph node data (A-K) **A.** Experimental design; WT or STAT6 KO (generation 25 or 24, respectively) adapted worms infected WT mice, and flow cytometry was run on cells from mesenteric lymph nodes at day 8 post infection. **B.** Gating strategy. **C.** Total number of mesenteric lymph node cells. **D-E.** Percent and total cell number of IL-13+ Th2 cells. **F-G.** Percent and total cell number of ST2+ Th2 cells. **H-I.** Percent and total cell number of IL-17+ γδT cells. **J-K.** Percent and total cell number of IFNγ+ CD8+ T cells in mesenteric lymph nodes. Data from the peritoneal cavity (L-T). **L.** Gating strategy to detect myeloid cell populations; cell types outlined by boxes. **M-N.** Percentage of parent population or total cell number of FcεR positive cells. **O-P.** Percentage of parent population or total cell number of macrophages. **Q-R.**

Percentage or total cell number of Arg-1 expressing macrophages. **S-T.** Percentage of parent population of eosinophils or neutrophils. Each point is a replicate mouse, and error bars are SEM. * $p < 0.05$ ** $p < 0.01$, *** $p < 0.001$, by t-test. Data are representative of 3 independent experiments.

granulocytes like neutrophils and eosinophils (Fig 6S and 6T). Upon analyzing the small intestinal tissue for changes in length (S6 Fig) and mRNA transcript levels of epithelial cell lineage markers for goblet cells (S6B–S6D Fig) or tuft cells (S6E Fig) using qPCR, there were no differences observed. This suggests some specificity to the helminth protective effector responses elicited by STAT6 KO-adapted worms. To address whether increased parasite burden was responsible for increased inflammatory responses against STAT6 KO-adapted *N. brasiliensis* infection, we ran multiple linear regression models comparing cell number to the day 8 adult parasite burden, AUC, or day 8 egg/g feces. We observed no correlation between any measure of parasite burden at the time of tissue harvest and immune response level, whereas prior host selection genotype significantly predicted the immune response level (S3 Raw Data). In addition, we did not find significant differences in adult parasite burden or egg shedding between WT- or STAT6 KO-adapted worms (S7 Fig), implying that the increased immune response is independent of parasite burden. Curiously, however, over the course of eight experiments we found that infection of WT mice with STAT6 KO-adapted worms led to a significant increase in host mortality (S8 Fig). We next tested whether a single generation of prior exposure to STAT6 KO hosts is sufficient to induce increased immunogenicity in *N. brasiliensis* by comparing the myeloid and lymphoid cells of WT mice infected with Parental *N. brasiliensis* that had been passaged for one generation in either WT or STAT6 KO mice. We found that the mesenteric lymph nodes did not differ in numbers and proportions of cell populations expressing cytokines IL-13, IL-17, or IFNγ, or the ST2 receptor between the STAT6 KO and WT conditions (S9B–S9J Fig). There was, furthermore, a lack of increase in any of the PEC myeloid cell populations previously measured (S9K–S9P Fig), while the parasite burden was similar (S9Q–S9S Fig), suggesting that more than a single generation of prior STAT6 KO host selection is required for increasing hookworm immunogenicity. Collectively, this suggests that multi-generational STAT6 KO-adapted worms have lost the ability to dampen the host inflammatory response compared to WT-adapted worms. This inflammation was not biased towards the Th2 immune response.

## WT and STAT6 KO-adapted hookworms have comparable reproductive fitness measures when challenged in WT hosts

Given that STAT6 KO-adapted hookworms appear to elicit a stronger host response, we hypothesized that such a response would negatively impact the overall fitness of worms if reintroduced to a wild-type host. Prior studies have found that while *N. brasiliensis* is a natural nematode of rats, it can adapt to better survive in mouse hosts following selection for as few as six generations [56,57]. We reasoned that adaptation to the STAT6 KO immunodeficient host for 6 to 24 generations would be maladaptive for their survival in a WT host. To test this, we challenged our adapted lines in WT mice, using a similar infection strategy as Fig 6A, and measured worm survival, fecundity, and transcriptional profile (Fig 7A). Contrary to expectation, we found that STAT6 KO-adapted worms had an egg-shedding pattern similar to WT-adapted worms from days 5–8 post-infection, which was confirmed *ex vivo* (Fig 7B–7D). We also found no evidence for decreased survival, with a similar adult parasite burden at day 9 (Fig 7E). These results were recapitulated multiple times between generations 6 to 24 (S10 Fig). We tested whether additional phenotypes were attenuated in STAT6 KO-adapted worms, and found no difference in worm length, hemoglobin, or ATP levels (Fig 7F–7H). The data

suggest that despite an increased WT host immune response, STAT6 KO adaptation does not confer any major disadvantage regarding survival or fertility of the adapted *N. brasiliensis* upon infection of a WT host. The increased length, ATP, and hemoglobin levels of *N. brasiliensis* observed within the STAT6 KO host did not persist even with one generation's exposure to the intact immune environment. This suggests that *N. brasiliensis* may undergo rapid transcriptional changes to adapt to a new host environment.

To test this, we sequenced transcripts of individual adult female worms from each cohort (generation 9 for previous WT-adaptation, or generation 7 for previous STAT6 KO-adaptation) after infection in a WT host. We found that prior STAT6 KO or WT adaptation does not result in separable groups by PCA or hierarchical clustering of the whole transcriptome expression when they encounter the WT host environment (Fig 7I and 7J), and only seven differentially expressed genes (Fig 7K). Interestingly, among the differentially expressed genes, three contained signal peptide motifs (S4 Raw Data), suggesting they may be released into the host environment. The most down-regulated gene encoded an astacin protein; astacins are highly expressed in the ES of adult stage worms [58], raising the possibility that these few transcriptional differences between WT- and STAT6 KO- adapted worms could impact the host. However gene set enrichment analysis revealed that, as a whole, expression of ESP-associated genes is not changed in STAT6 KO-adapted worms (S11 Fig). To better visualize differences between conditions, three outlier samples were excluded from the analysis based on being outside of the 95-inner-percentile range of PC1 and PC2, and having strong influence on row-z-score distribution in the heatmaps (S12 Fig). Their exclusion does not impact our overall conclusions. These data indicate that despite the dramatic change in transcriptome and fitness phenotypes of STAT6 KO-adapted *N. brasiliensis* within their passage hosts, within 7–9 generations such worms revert to a transcriptional state likely suited for a WT host upon re-exposure to this immunocompetent environment.

## Discussion

This study investigated whether a host environment lacking STAT6-dependent immune responses could negatively affect the fitness of hookworm parasites once they were reintroduced to a host with an intact immune environment. Our results collectively suggest that although STAT6 immunity pressures hookworms to adaptively dampen their immunogenicity, hookworms can rapidly restore most biological features that were previously lost over the span of a single generation in an immunocompetent host. Host adaptation in helminth species is known to occur, but whether specific arms of host immunity are responsible has not been addressed. This work reveals that loss of STAT6-driven immune responses increases hookworm size and reproductive output and significantly alters the transcriptional profile of these parasites as early as day 5 post-infection. Curiously however, it takes multiple generations of adaptation to the STAT6 KO environment before diminished expression of ESP-associated genes can be observed, suggesting that hookworms continue to change over repeated exposure to an immunodeficient host. Interestingly, our data show that hookworm gene expression is dynamically lowered upon initial residence in STAT6 KO hosts, but multi-generational exposure leads to downregulation of genes more associated with host immune modulation. Accordingly, challenge of an immunocompetent host with STAT6 KO-adapted worms elicits a greater inflammatory response than normal (Fig 8). Even so, STAT6 KO-adapted worms rapidly regain most of their WT-host transcriptional profile when reintroduced into immunocompetent hosts for a single generation. Phenotypic traits like size and fecundity that increase in the absence of STAT6 are quickly reversed in the presence of type 2 immunity (Fig 8). Collectively, these data highlight the remarkably versatile adaptation of hookworms to their hosts,

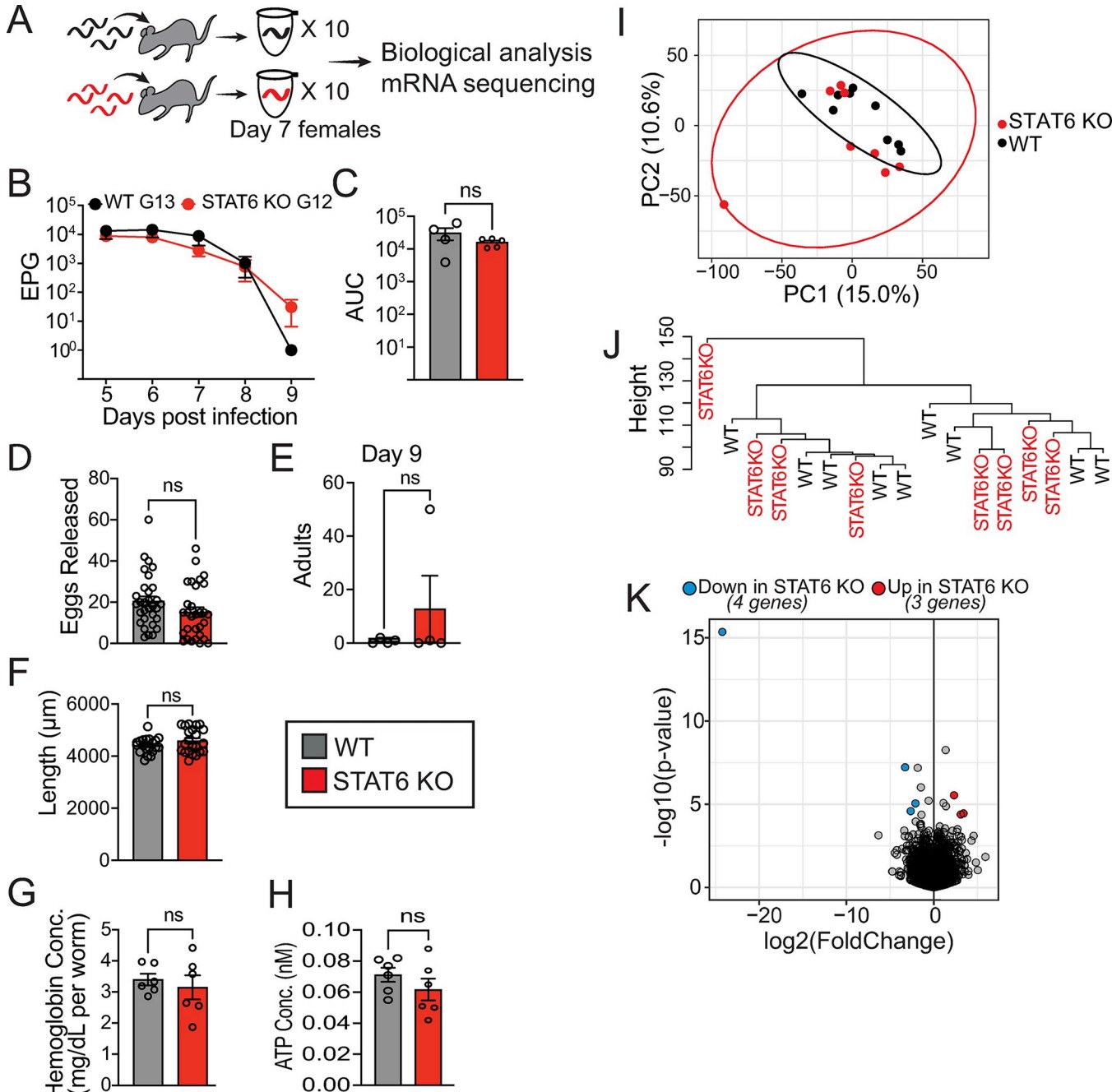

**Fig 7. STAT6 KO-adapted *N. brasiliensis* do not exhibit attenuation of fitness when challenged within WT mice. A.** Experimental design. WT (black) or STAT6 KO (red) adapted worms were isolated from WT hosts at day 7 post infection for RNA sequencing or indicated time points for parasitology or morphological measurements. Individual female worms were used for mRNA sequencing. **B.** Mean and SEM of eggs per gram of feces (EPG) by days post infection for generation-13 WT mouse-adapted *N. brasiliensis* (WT G13) and generation 12 STAT6 KO mouse-adapted *N. brasiliensis* (STAT6 G12) infection of WT mice; 4 replicates per condition. **C.** AUC of B by host-adaptation type. **D.** Eggs released from individual adult worms after a 25 hour *ex vivo* incubation in liquid media, recovered from the WT host at day 8 (G25/24 adapted lines). **E.** Adult worms from infections with adapted *N. brasiliensis* recovered at day 9 post infection from the intestines of WT hosts. **F.** Worm length for each adapted *N. brasiliensis* (G25/24) after WT exposure. **G.** Worm-hemoglobin concentration for WT-mouse infection by passage condition (G15/14 adapted lines) at day 8 post infection. **H.** Worm-ATP concentration from WT-derived adult day 8 post infection, adapted (G24/25) *N. brasiliensis*. For C-H plotted data includes individual values (circles), means (bars), SEM (error bars), ns p-value > 0.05 by t-test. Data is representative of 8 (Fig 7B and 7C) or 2 (Fig 6D–6H) independent experiments. **I-K.** RNA-seq data of individual day 7 post infection WT or STAT6 KO (G9/7) adapted *N. brasiliensis* derived from WT host infection. **I.** Principal component analysis (PCA) plot of principal components (PC) 1 and PC2, for individual worm gene expression with ellipses around the 95% normal probability for each host adaptation condition. **J.** Hierarchical clustering of gene expression data. **K.** Volcano plot of differential gene expression results; significance indicated by Log2FC > 2 or < -2, and adjusted p-value < 0.05.

which could be an alternative mechanism for development of drug resistance for parasitic nematodes in addition to known mutations in genomic loci [59–62].

Adult worms in the intestines are located in a microenvironment subject to immune-mediated expulsion and inhibition of feeding activity, which is distinct from L4 in the lung environment that are subject to larval killing and trapping. Key STAT6-driven lung immune responses include trapping of larvae by M2 macrophages, whereas gut effector responses largely depend on epithelial cell effector molecules [29,63]. Thus, there are clear differences in host selective pressure depending upon the different life cycle stages. By this work and others, it is evident that the parasite-intrinsic factors that counter the immune host immune response are quite different between the lung and gut stage [58]. Prior findings, and our data, indicate that STAT6- and IL-4R-driven immunity impacts gut-resident adult stage parasites, but not lung-resident larval stage worms (S1 Fig) [30]. Our RNA-seq data further support this hypothesis, showing that L4 stage larvae from STAT6 KO or WT hosts are transcriptionally indistinguishable. This is consistent with minimal effects of STAT6-dependent responses on lung parasites. Moreover, the short duration of time that larvae spend in the lung microenvironment, suggests that the parasites do not have sufficient time to sense and adaptively respond to potential differences in immune lung immune responses between WT and STAT6 deficient hosts. Unlike transient migration of larvae through the lung, mature hookworm species reside in the intestines for several days, during which diecious adults sexually reproduce and shed eggs. Adult worms likely secrete more immunomodulatory compounds than larval stages, given that the former has a longer residence in one organ system and has a larger known repertoire of ESPs [58]. A prior study in another helminth species, *S. ratti*, found that selection of progeny from late-infection-stage worms survived longer when placed in the "hostile" environment of an immunized host compared to progeny selected at an earlier age of infection [15]. The broad implication of this study is that worm survival traits are constantly under host selection pressure exerted by immunity; however, it does not rule out other factors that influence immune responses (e.g., microbiota, metabolic state, nutritional status). Our present work that employs a genetically defined host immune environment shows how it impacts parasite morphology, physiology, fecundity, and transcriptional regulation.

STAT6 is one of the canonical transcription factors necessary for establishing and maintaining type 2 immune dominance during helminth infection [28,64,65]. STAT6 activation occurs downstream of either type 1 or type 2 IL-4 receptors that bind IL-4 or IL-4/IL-13, respectively [66]. Numerous studies have demonstrated that loss of this signaling pathway in hosts leads to higher hookworm burdens in the GI tract [27,30]. The impact of type 2 immunity on adult hookworm reproductive fitness in the GI tract might be due to goblet and tuft cell expansion, which are both dependent upon STAT6-driven immunity. Goblet cells in the GI tract secrete mucin, which is thought to help dislodge worms from their feeding niche. In addition, goblet cell-derived factors like RELMβ are thought to inhibit feeding behaviors by directly binding the worms [31]. Transcriptionally, worms in WT hosts express more heat shock proteins, which suggest a stress response [67] and neuropeptides, which suggest evasion of potentially noxious immune factors [50]. Our phenotypic and transcriptomic findings support the idea that STAT6-driven immunity not only expels worms from their niche but also diminishes the size, feeding capacity, and fecundity of individual adult female worms. Chemosensory tuft cells that activate ILC2 in the GI tract are also highly expanded during a type 2 immune response in a STAT6- or IL-4R-dependent manner [68–70]. These specialized epithelial cells are responsible for sensing the presence of worms, perhaps by detecting worm-produced metabolites [71]. Our study found that after selection in the STAT6 KO environment, hookworms have reduced transcription of gene families observed in ESPs, while eliciting a stronger immune response from WT hosts. Interestingly, this increased WT host immune

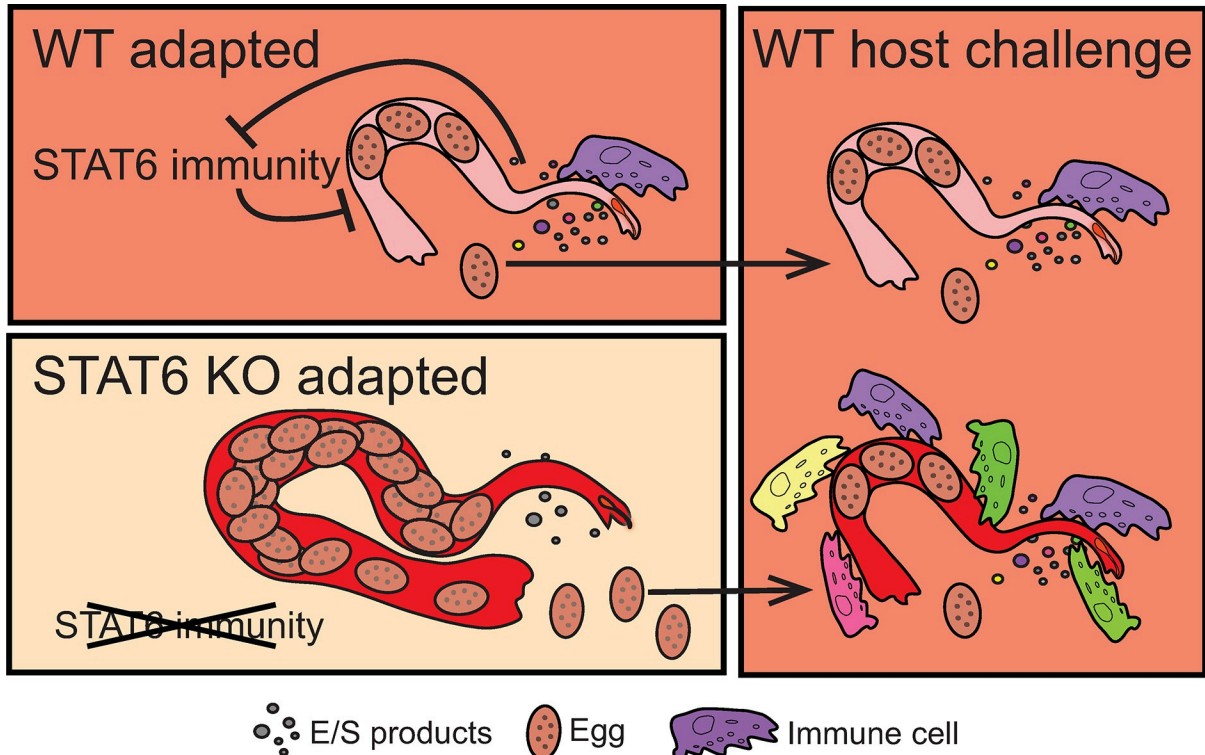

**Fig 8. Graphical summary.** WT-adapted hookworms within WT host environment (upper left box) have impeded survival and fecundity due to STAT6 immune responses. In response, worms produce excreted and secreted products (E/S), which may dampen STAT6 driven immunity. Worms adapted to hosts lacking STAT6 immunity (bottom left box) are larger, shed more eggs, and have decreased E/S production. When progeny of adapted worms infect WT hosts (right box), they revert to similar size, egg-shedding, and production of E/S. However, progeny from STAT6 KO adapted worms elicit a stronger immune response within WT hosts.

response did not manifest increased tuft cells or goblet cells and the fitness of the worms was also not apparently impacted. Given that protective immunity to hookworms hinges on tuft cell expansion and activity that drives protective ILC2 responses [69], the diminished expression of ESP-associated gene categories in STAT6 KO-adapted hookworms may be a consequence of reduced tuft cell responses. Ongoing experiments seek to further characterize the difference in ESPs of WT- versus STAT6 KO-adapted worms via mass spectrometry and address whether these divergent ESP profiles have altered immunogenicity.

A hallmark of Type 2 immunity is minimizing the production of Type 1 inflammatory responses that are harmful to the host if left unchecked [16,72]. Evidence suggests that both tissue damage and hookworm excreted/secreted products are required to elicit an appropriate type 2 immune response [73], while excreted/secreted products alone can promote anti-inflammatory and immunosuppressive responses [54,55]. Mechanisms for dampening inflammation are diverse, and include promotion of Treg proliferation [74], direct inhibition of neutrophils [75] and macrophages [76], antagonizing host lectins during feeding [77], and through miRNA silencing [54]. Our findings are consistent with an altered ability to suppress the sensing and/or amplification of type 2 immunity, or perhaps, the differentiation of M2 macrophages. Our data suggest that the immune state of prior generations can determine the inflammatory response that hookworms elicit from their host. Moreover, our results imply that hookworms somehow actively sense the host's immune response and adjust ESP-associated gene expression in a manner tuned by prior generations' exposure to host immunity.

Interestingly, our WT-host-challenge data was somewhat inconsistent with prior studies who found that other parasitic helminths become more fit after immune selection [15,78]. These findings are puzzling as one would expect a stronger immune response would weaken at least the survival, if not the fecundity, of parasites. Indeed, we find increased host mortality with STAT6 KO-adapted worm infection (S8 Fig). Given that excessive immune activity can harm the host, which negatively impacts the parasite's fitness if host death occurs, our results are consistent with selection for immune-dampening traits that are host-protective.

Our gene expression data showed a dramatic difference between WT and STAT6 KO-adapted worms within their passage hosts, which largely went away when the 7th-generation STAT6 KO worm stock was reintroduced to the WT host environment. After reintroduction, we found that STAT6 KO-adapted worms were globally indistinguishable from WT, with previously suppressed pathways restored. We also found that the parental (WT rat-maintained) stock of *N. brasiliensis* dramatically altered its transcriptional profile upon exposure to STAT6 KO hosts (G1 infection). This suggests that most observed transcriptional differences were transient and host-environment specific. However, to conclude that this specific transcriptional profile suits the immune deficient versus sufficient environment would require further experimentation, such as examining the gene identity and expression magnitude in worms exposed to hosts with enhanced STAT6 driven immunity, like in a secondary infection challenge. This finding is somewhat incongruous with the observation that STAT6 KO-adapted worms elicit a stronger WT host immune response after 25 generations of selection, as one might expect a more dramatic shift in gene expression profile within WT hosts. However, the relevant transcriptional differences might have only arisen after generation 7 or occurred in an earlier developmental stage; alternatively, the observed enhanced host immune response might be due to altered expression of only a few genes in STAT6 KO-adapted worms rather than extensive gene families. Indeed, we found a large decrease in the expression of an Astacin-metallopeptidase protein in the STAT6 KO-adapted worms, which may suggest a novel immune-dampening function of this abundantly secreted protein [79].

Some limitations of our study are as follows. RNA-sequencing of initial (G1) versus adapted (G9/7, 15/14) was done within different institutions; hence, factors like animal facilities, parasite stocks, and sequencing batch effects could have influenced the differences observed. Second, our adapted cohorts were not done with experimental replicates. In addition, the number of worms passaged with each generation was between 1,000–1,500 infectious larvae spread across 2–3 hosts; hence, differences we are attributing to immune selective pressure may have instead arisen due to bottleneck or other random effects. Lastly, while it is well established that *N. brasiliensis* maintained in WT rats has increased burden within STAT6 KO hosts, we only tested additional parasite measurements (such as size, ATP content, and hemoglobin content) after they had been exposed to STAT6 KO environments for many generations; therefore, we do not know how early during selection these features appear, although we have provided the rationale for conducting the study in the manner performed.

Our results further substantiate the unique adaptability of mammalian hookworms and may support a worm-adaptation based mechanism for human hookworm burden increasing with age. In identifying host-immune-altered pathways like neuropeptide signaling, proteases, SCP/TAPS proteins, and transthyretin-like proteins, this study provides insight into the most dynamically regulated mechanisms in response to a loss of host immunity. Future studies will explore whether passage of hookworms in hosts with enhanced STAT6-driven immunity (i.e., through host treatment with exogenous IL-4) further increases ESP expression above that of WT hosts. Identifying such ESPs could provide important targets for drugs or vaccines against adult-stage worms and will lay the foundation for further study into the mechanism of adaptation in parasitic nematodes.

## Materials and methods

### Ethics statement

All procedures involving animals were approved by the University of Pennsylvania institutional animal care and use committee (IACUC) (protocol 805911) and conformed to American Veterinary Medical Association (AVMA) guidelines. Mice were maintained at the University of California, Riverside in a similar barrier facility. All procedures involving animals were approved by the University of California, Riverside IACUC (protocol 20210017) and conformed to AVMA guidelines.

### Mouse strains

Mouse strains used for passage of *N. brasiliensis* and measurements of *N. brasiliensis* parasitology features were maintained at the University of Pennsylvania. B6.129S2(C)-Stat6$^{tm1Gru/J}$ mice (STAT6 KO), and wild type C57BL/6J mice (WT) were obtained from The Jackson Laboratory bred in our colony, and housed in a barrier facility free from other pathogens. Female Rats were purchased from Wistar and were housed in a separate room of the same facility. Mice originating from the same Jackson laboratory strains as above were used for sequencing of single generation (G1) infected WT or STAT6 KO mice.

### *N. brasiliensis* serial passaging and maintenance

*N. brasiliensis* maintained in wild-type Wistar rats were used to infect, via subcutaneous injection of 500 infectious larval stage worms (iL3), 3 WT or 3 STAT6 KO mice to generate the first and subsequent generations of passaged mice. WT passaged worms were initiated one generation in advance of STAT6 KO-passaged worms. Worm cultures were made from the feces of either infected WT or STAT6 KO mice collected overnight from day 6 to day 7 post-infection, with single culture plates made for either group. Cultures were made with a mixture of charcoal and peat moss, as described previously [18]. Subsequent passages were made from iL3 of 2–3 week-old worm culture plates into 2–5 hosts, and passage-host sex was alternated between males and females. Passage hosts were 5–10 weeks old at the time of infection. Passage of G1 worms at UCR was in female Sprague-Dawley rats purchased from Taconic, following the same methodologies as above.

### Adult-stage worm assays

Adult-stage worms were isolated from individual host small intestine as previously described [8] by dissecting the small intestines from duodenum to ileum, cutting longitudinally, and placing the tissue on a modified Baerman apparatus lined with two layers of gauze over a beaker of PBS, and incubating for 2–3 hours at 37˚C. Worms settled at the bottom of the beaker were transferred to petri dishes and transferred individually under a dissecting microscope by pipette for assessment. Worm sex was determined visually by distinguishing morphological features [18]: individuals with a fan-like tail and smaller body size were mature males, while those with largest body size and darker pigmentation and a pointed hook-like tail were mature females. Worm hemoglobin content was measured by isolating 10 individual adult-stage males or females per measurement into 1.5 mL Eppendorf tubes, washing worms once with deionized water and three times with PBS, leaving a 50 μL volume. Worms were homogenized with a plastic mortar and immediately assessed for hemoglobin concentration using the colorimetric Hemoglobin assay kit (Sigma-Aldrich, MAK115) according to manufacturer's recommendations. Worm ATP content was measured as previously described [63], using 10 worms per measurement (live or heat-killed by boiling for 10 minutes).

For imaging, live adult worms were isolated, washed 3X in PBS, resuspended in 25 mM sodium azide PBS, and mounted on glass slides with agarose pads as previously described [80]. Worms were imaged at 10X magnification on a Leica DM6000B widefield microscope with a Leica DMC-2900 color camera. For worms that did not fit within a single image field, the image tiling and stitching function was used to capture the full length of the worm. Images were acquired as Z-stacks and processed with the extended depth of field function within the Leica LAS X software. To measure worm length, images were exported as tiff images with scale bars, which were used to create an appropriate pixel-to-micron conversion calibration within MetaMorph image analysis software. Length was measured by tracing worms with the calibrated line tool within MetaMorph and approximating curves with a series of connected short straight-line segments.

Excreted and secreted products were isolated from adult stage worms as follows. Worms were isolated from the small intestine of WT or STAT6 KO hosts after 25 or 24 generations of selection at day 7 post infection 500–1000 mixed sex worms were isolated into PBS, washed in sterile PBS 5X, then incubated in serum-free RPMI with Penicillin-Streptomycin (1 mg/mL) and gentamycin (0.5 mg/mL) and 10% fungizone for 30 minutes at 37˚C, then washed 3 times with worm media (RPMI with 1% glucose, Penicillin-Streptomycin (100 μg/mL), gentamicin (50 μg/mL) and 1% fungizone). Worms were incubated at 37˚C, discarding and replacing half of the supernatant after 25 hours, and collecting media supernatant after 3 days. Protein was quantified using a BCA assay, and 10 μg/mL was used to stimulate cells.

## mRNA sequencing of *N. brasiliensis*

For WT- or STAT6-KO adapted *N. brasiliensis* (G9/7) from WT challenge infection or G15/14 from passage hosts), RNA isolation and sequencing were carried out at the University of Pennsylvania, Department of Pathobiology Facilities, as follows. Adult-stage worms were isolated from single hosts for each condition and washed five times with sterile PBS. Individual worms were transferred by pipette into sterile tubes and snap-frozen. Isolates were then thawed and homogenized with a pestle. RNA extraction was performed with a trizol phenol-chloroform method. RNA integrity was analyzed by high-sensitivity RNA TapeStation (Agilent, 5067–5579), samples with RIN values of 7 or higher were included for sequencing. RNA concentration was determined by qubit (Invitrogen, Q32852). Libraries were made from 1 ng RNA per sample with the mRNA SMART-seq HT kit (Takara, 634437). Libraries were sequenced using the Nextseq 550 (Illumina) with single-read 75 cycles for WT challenge RNA-seq (Fig 7), or the NextSeq 2000 (Illumina), single-read 100 cycles for within passage host RNA-seq (Figs 3 and 4). At least 20 million raw reads were obtained per sample.

For single generation (G1) infections of WT or STAT6 KO mice, parasites were harvested at the University of California, Riverside, from 3 hosts per condition. Individual lung-stage L4 worms were isolated as previously described [81]. Individual D5 and D7 adult-stage female worms were isolated by dissecting the entire small intestine of the host and placing it in a petri dish with PBS. After approximately two hours of incubation at 37 degrees, adult worms were manually picked out of the intestinal tissue, evaluated for sex, and placed in PBS.

Worm sex was determined based on morphological features under a 5x dissecting microscope; females are distinguished from males by the presence of the vulva and eggs and the absence of a copulatory bursa. RNA from individual worms was then isolated as previously described [82]. mRNA libraries were prepared using a Nextera DNA Flex Library Prep Kit (Illumina, #20018704) and sequenced on an NovaSeq 6000, using S1 50 bp, paired-end reads, to a depth of 10–60 million paired reads per sample.

## mRNA-seq analysis

For RNA-seq analyses, we used a protein-coding gene set from an improved genome assembly of *N. brasiliensis* strain MIMR (GenBank accession GCA_030553155.1). Nanopore sequencing reads were assembled with raven [83]; contigs were error-corrected with Nanopore reads and racon [84], then with Illumina reads and POLCA [85]; chromosomes were scaffolded with Omni-C reads and 3d-dna [86]; protein-coding genes were predicted with BRAKER2 [87] and TSEBRA [88]; protein-coding genes were annotated with InterProScan [89] and EnTAP [90]. Full biological analysis of this genome will be published elsewhere.

Raw sequencing reads were trimmed to remove low-quality or adapter-containing sequences using Trimmomatic v. 0.39 [91], with the following settings LEADING:3 TRAILING:3 SLIDINGWINDOW:4:15 MINLEN:36. K-mers from trimmed reads were then aligned to coding DNA sequences (CDSes) derived from the *N. brasiliensis* newly assembled reference genome (PRJNA994163, GCA_030553155.1) using the kallisto k-mer pseudoaligner, with the following settings for G15/14 and G9/7 infections: -b 100—single -l 100, and the following settings for G1 infections: -b 100 [92]. Pseudocount data was imported to R using tximport [93], filtered to retain genes with expression values greater than 0 for 5 or more samples, normalized and scaled to obtain counts per million (CPM) values using edgeR [94], which were used for principal component analysis performed using the prcomp function and plotted using ggbiplot in R [95]. Differentially expressed genes were identified from raw count data using DESeq2 [96], and the DESeqDataSetFromTximport function. Heatmaps based on row-z-scores of CPM values were plotted with the heatmap2 R package, and volcano plots were generated with ggplot. Gene set enrichment analysis (GSEA) was performed using the GSEA software on CPM values [42,43]. Gene identities of peptides previously identified from adult stage *N. brasiliensis* secreted in culture [58] were identified using a BlastP search of peptide sequences against predicted *N. brasiliensis* protein sequences [97,98].

## Host immune analysis

Cells from WT mice infected with WT or STAT6 KO-adapted *N. brasiliensis* were isolated from either mesenteric lymph nodes (MLNs), or peritoneal exudate (PECs) fluid washed with 5–7 mL of sterile PBS. MLNs were mechanically disrupted, filtered, and cell suspensions were generated. PECs were stained to evaluate myeloid cell populations using the antibodies listed in S1 Table. MLN cells were stimulated with Cell Activation cocktail with Brefeldin A (Biolegend, Cat#. 423304) for 5–6 hours, followed by live dead staining using the LIVE/DEAD Fixable Aqua Dead Cell Stain Kit, for 405 nm excitation (ThermoFisher, Cat#. L34966), surface stained for lymphocyte populations, followed by fixation and permeabilization with eBioscience Foxp3 / Transcription Factor Staining Buffer Set (ThermoFisher, Cat# 00-5523-00). Next, intracellular staining for cytokine production was performed using the antibodies listed in S1 Table. Myeloid cells were not stimulated prior to surface staining. Samples were acquired using a Symphony A3 Lite (BD Biosciences) and analyzed using Flowjo software. $6 \times 10^6$ cells from MLNs were also incubated with CD3/CD28 antibody for 48 hours, and supernatant was collected for detection of cytokines by ELISA (Invitrogen).

## Statistical analysis

Except where indicated, all plots and statistical tests were run in Graphpad Prism (v9). Data were normally distributed so parametric statistical tests were used, with post-hoc testing for more than two groups. Significance was set at $p < 0.05$.

## Supporting information

**S1 Fig. Early infection worm burden of WT versus STAT6 KO mice infected with WT rat maintained parental *N. brasiliensis*. A.** Experimental design. **B.** Number of parasites in the lungs after 24 hours, 48 hours, or 5 days post infection. **C.** Number of parasites in the gut after 48 hours or 5 days post infection. Each point is a replicate mouse, and error bars are SEM. ns p-value > 0.05, by t-test. Data are representative of 1 independent experiment.
(TIF)

**S2 Fig. Comparison of differentially expressed genes in adapted (G15/14) vs G1 adult female *N. brasiliensis* from STAT6 vs WT hosts. A.** Venn diagrams of significantly up (Log2FoldChange > 2) versus down (Log2FoldChange < -2) regulated genes for STAT6 KO versus WT host conditions (for both, adjusted p < 0.05). **B.** Scatterplot of all STAT6 KO vs. WT host condition Log2(FoldChange) values, filtered on adjusted p-value < 0.05, in single generation *N. brasiliensis* infection, (G1) (y-axis) versus adapted *N. brasiliensis* (G15/14) (x-axis). Values represent the number of genes in each quadrant.
(TIF)

**S3 Fig.** Heatmaps of gene sets of interest from **A.** Day 5 females WT versus STAT6 KO hosts infected with progeny of parental *N. brasiliensis* (G1), **B.** Day 7 females of G1, or **C.** Day 7 females worms from WT or STAT6 KO hosts infected with G15 or 14 adapted *N. brasiliensis* respectively.
(TIF)

**S4 Fig. Representative gating plots for flow cytometry.** A. Gating strategy to detect lymphoid cell populations from peritoneal cells. B. Gating strategy for detection of myeloid cell populations from mesenteric lymph nodes.
(TIF)

**S5 Fig. Excreted products from adult STAT6 KO -adapted worms have decreased immunogenicity.** Supernatant from 3-day cultures of WT or STAT6 KO- adapted (G24/23) *N. brasiliensis* was used to stimulate mesenteric lymph node cells collected from WT mice infected with either WT or STAT6 KO -adapted (G25/24) *N. brasiliensis* at day 8 post infection. A. concentration of IL-4 in supernatant. B. concentration of IL-10 in supernatant. P-values are from paired t-tests. C. Concentration of IFNγ measured following anti-CD3/28 stimulation for each infected host (circles), with SEM error bars. ***p-value < 0.001 by t-test. Data is representative of 2 independent experiments.
(TIF)

**S6 Fig. Intestinal measurements of WT mice following infection with WT- or STAT6 KO-adapted *N. brasiliensis*. A**. Length of intestine. **B-E**. RT-qPCR expression relative to GAPDH. Circles are individual infected hosts, bars are mean values, ns p-value > 0.05 by t-test. Data is representative of 1 independent experiment.
(TIF)

**S7 Fig. Parasitology associated with WT mice infected with WT or STAT6 KO adapted *N. brasiliensis* (G25/24) Fig 6. A.** Average eggs per gram of feces by days post infection, with SEM error bars. **B.** Average area under the curve of data plotted in A. **C.** Number of adult parasites in the small intestine at day 8 post infection.
(TIF)

**S8 Fig. WT host survival decreased with STAT6 KO-adapted *N. brasiliensis* infection. A.** Survival curve with average host survival by days post infection for WT hosts infected with

WT-adapted or STAT6 KO-adapted worms. Combined data from 9 independent experiments, 49 mice per condition, Mantel-Cox test p-value.
(TIF)

**S9 Fig.** *N. brasiliensis* **exposed to STAT6 KO hosts for one prior generation do not elicit a stronger inflammatory WT host immune response.** Mesenteric lymph node data (A-J) **A.** Experimental design; Parental worms infected WT or STAT6 KO mice, and resulting progeny (G1 adapted) infected WT mice. Flow cytometry was run on cells from mesenteric lymph nodes at day 8 post infection. **B.** Total number of mesenteric lymph node cells. **C-D.** Percent and total cell number of IL-13+ Th2 cells. **E-F.** Percent and total cell number of ST2+ Th2 cells. **G-H.** Percent and total cell number of IL-17+ $\gamma\delta$T cells. **I-J.** Percent and total cell number of IFN$\gamma$+ CD8+ T cells in mesenteric lymph nodes. Data from the peritoneal cavity (K-P). **K-L.** Percentage of parent population or total cell number of macrophages. **M-N.** Percentage or total cell number of Arg-1 expressing macrophages. **O-P.** Percentage of parent population of eosinophils and neutrophils. **Q.** Parasite burden by number of eggs per gram of feces (EPG) for each day post infection. **R.** Area under the curve. **S.** Number of worms from host small intestine at day 8 post infection. Each point is a replicate mouse, and error bars are SEM. * $p < 0.05$ ** $p < 0.01$, *** $p < 0.001$, by t-test. Data are representative of 1 independent experiment.
(TIF)

**S10 Fig. Summary of all WT challenge experiments with either WT or STAT6 KO adapted worms. A-C.** Plots of eggs per gram of feces (EPG) by days post infection of WT mice infected with STAT6 KO- or WT-adapted *N. brasiliensis* percentage of surviving adults from the initial inoculum number shown below. Number of generations of adaptation and sex of mice noted as generations in STAT6 KO/WT mice above plots. Data collected until day 9 (**A**), day 8 (**B**), or day 7 (**C**) post infection.
(TIF)

**S11 Fig. Heatmaps of gene expression data from WT or STAT6 KO (G 9/7) adapted female worms derived from the intestine at day 7 post infection of WT mice. A.** Heatmap and GSEA enrichment plot for genes encoding adult stage ESP-associated gene categories. **B.** Heatmaps for expression of genes in Chromosome, Gene Silencing, or FMRFamide-like families.
(TIF)

**S12 Fig. Gene expression data from WT or STAT6 KO (G 9/7) adapted female worms derived from the intestine at day 7 post infection of WT mice, with outlier samples included (indicated by \*). A.** PCA plot. **B.** Hierarchical clustering. **C.** Volcano plot of differential gene expression results; significance indicated by Log2FC > 2 or < -2, and adjusted p-value < 0.05. **D.** Heatmap and GSEA enrichment plot for genes encoding adult stage ESP-associated gene categories. **E.** Heatmaps for expression of genes in Chromosome, Gene Silencing, or FMRFamide-like families.
(TIF)

**S1 Table. Extended list of key reagents.**
(DOCX)

**S1 Raw Data. Complete GSEA results in G1 day 7 adult worms from WT or STAT6 KO mice.**
(XLSX)

**S2 Raw Data. Complete GSEA results in G15/14 adapted day 7 adult worms from WT or STAT6 KO mice.**
(XLSX)

**S3 Raw Data. Summary of regression analysis of host immune cell populations versus hookworm adaptation genotypes, and hookworm burden, in WT mice infected with WT or STAT6 KO G15/14 -adapted *N. brasiliensis*.**
(XLSX)

**S4 Raw Data. Complete GSEA results in G1 day 5 adult worms from WT or STAT6 KO mice.**
(XLSX)

## Acknowledgments

We thank Drs. Boris Streipen and Dustin Brisson for their input regarding interpretation of findings in generational experiments, and Dr. Gordon Ruthel of the Penn Vet Imaging Core for guidance with imaging and size analysis of *N. brasiliensis* worm lines. We also thank Dr. Daniel Beiting and Clara Malekshahi for the use of their sequencing facility and Linux computing cluster.

WEHI acknowledges the Victorian State Government Operational Infrastructure Support and the Australian Government National Health and Medical Research Council Independent Research Institute Infrastructure Support Schemes.

## Author Contributions

**Conceptualization:** Erich M. Schwarz, Meera G. Nair, Adler R. Dillman, De'Broski R. Herbert.

**Data curation:** Annabel A. Ferguson, Erich M. Schwarz.

**Formal analysis:** Annabel A. Ferguson, Juan M. Inclan-Rico, Dihong Lu, Sarah D. Bobardt, LiYin Hung, Quentin Gouil, Louise Baker, Matthew E. Ritchie, Aaron R. Jex, Erich M. Schwarz.

**Funding acquisition:** Erich M. Schwarz, Meera G. Nair, Adler R. Dillman, De'Broski R. Herbert.

**Investigation:** Annabel A. Ferguson, Juan M. Inclan-Rico, Dihong Lu, Sarah D. Bobardt, LiYin Hung, Quentin Gouil, Louise Baker, Matthew E. Ritchie, Aaron R. Jex, Erich M. Schwarz.

**Methodology:** Annabel A. Ferguson, Juan M. Inclan-Rico, LiYin Hung, De'Broski R. Herbert.

**Project administration:** De'Broski R. Herbert.

**Supervision:** Erich M. Schwarz, Meera G. Nair, Adler R. Dillman, De'Broski R. Herbert.

**Validation:** Annabel A. Ferguson.

**Visualization:** Annabel A. Ferguson, Heather L. Rossi.

**Writing – original draft:** Annabel A. Ferguson, De'Broski R. Herbert.

**Writing – review & editing:** Annabel A. Ferguson, Juan M. Inclan-Rico, Sarah D. Bobardt, Aaron R. Jex, Erich M. Schwarz, Heather L. Rossi, Meera G. Nair, Adler R. Dillman, De'Broski R. Herbert.

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
