## [Decision Letter · Decision Letter 0]

26 Jun 2023

Dear Dr. Herbert,

Thank you very much for submitting your manuscript "Hookworms dynamically respond to loss of Type 2 immune pressure" for consideration at PLOS Pathogens. As with all papers reviewed by the journal, your manuscript was reviewed by members of the editorial board and by several independent reviewers. In light of the reviews (below this email), we would like to invite the resubmission of a significantly-revised version that takes into account the reviewers' comments.

Most importantly, please pay attention to the proposed additional work suggested related to secondary infection/protection in WT mice. 

We cannot make any decision about publication until we have seen the revised manuscript and your response to the reviewers' comments. Your revised manuscript is also likely to be sent to reviewers for further evaluation.

Sincerely,

Thomas B. Nutman

Academic Editor

PLOS Pathogens

James Collins III

Section Editor

PLOS Pathogens

Kasturi Haldar

Editor-in-Chief

PLOS Pathogens

orcid.org/0000-0001-5065-158X

Michael Malim

Editor-in-Chief

PLOS Pathogens

orcid.org/0000-0002-7699-2064

Reviewer's Responses to Questions

**Part I - Summary**

Reviewer #1: This study investigates transcriptional changes in N. brasiliensis when passaged in immunodeficient mice (STAT6KO), to assess how immune pressure affects parasite responses. The authors show that N. brasiliensis can rapidly alter its transcriptional programme in STAT6 KO. These changes do not affect parasite viability in vivo, however they do increase the size of the immune response against the parasite. On infection of STAT6KO-adapted parasites into WT mice, the transcriptional profile of parasites returned to indistinguishable to control parasites.

This studies strengths are around its approach to the experimental system, to see how immune responses affect parasite transcription. Its weaknesses are well addressed in the text, and include bottlenecks of parasite numbers and the effect of immune pressure on proteomic, as well as transcriptional responses of the parasites. However the approach is reasonable and robust, and future studies can address these other issues.

Reviewer #2: The manuscript PPATHOGENS-D-23-00743 details that adaptation of N. braziliensis under decreased immunological pressure in STAT6 KO animals. Ferguson et al. elegantly show that Nb maintained in STAT6 KO mice for up to 25 generations shows increased body size and higher egg production. In addition, several transcriptional changes were observed in agreement with lower immunologic pressure on the parasites, including decreased expression of genes involved in motility and excreted and secreted products, that modulate host’s immune system. Increased transcription pathways involved genes associated with reproduction and physiological fitness. Interestingly, these adaptations did not translate to inherited features and the first infection in WT animals, STAT6 KO-adapted parasites returned mostly to their original transcriptional landscape with few exceptions. On the other hand, it seems that adapted parasites lost their ability to downmodulate the immune response, consistently inducing more robust inflammation, which did not translate to fewer viable adult parasites in the host’s gut. This is an sophisticated and well-designed work which brings important information on host-parasite interaction and how immunological pressure can shape a parasite adaptation.

Reviewer #3: This work performed by Ferguson and collaborators investigated whether serial passage of the rodent hookworm Nippostrongylus brasiliensis in STAT6-deficient mice caused changes in parasites over time. By using an improved N. brasiliensis genome, the authors found that these physiological changes observed in the STAT-6 KO-adapted strains corresponded with a dramatic shift in the transcriptional profile, including increased expression of gene pathways associated with parasite persistence. This is interesting story of potential interest to the field, however, there are several issues that need to be addressed to improve the paper, including the fact that the current findings do not sufficient to support the conclusions.

**Part II – Major Issues: Key Experiments Required for Acceptance**

Reviewer #1: N/A

Reviewer #2: (No Response)

Reviewer #3: Major comments:

1) The findings presented on this current manuscript don’t show enough evidence to suggest that hookworms alter their transcriptional profile to suit their immediate host immune environment. The most remarkable result is that there is a group of two thousand genes that are downregulated in the adult worms from a parental strain that established the infection in the absence of a type-2 immunity. However, to address the question whether the alteration in those genes were to suit the pressure of the immune response, the authors should have followed up this observation and performed another assay where parental or adapted strains are used in a protective secondary model in WT mice. The analysis of the transcriptional profile of the worms under the pressure of a protective type-2 immune response in comparison with the molecular program of the ones obtained from a secondary model in the STAT-6 KO mice would be very important to associate with those 2,000 genes downregulated in the primary infection.

2) As shown in figure 3C, the infection with the parental (rat-adapted) strains into STAT-6 KO mice followed by transcriptional analysis in the G1 parasites revealed the largest differences in their molecular profile when compared with parental strain-infected WT mice. This change was also relevant even when compared with the STAT-6-adapted G14 strain. With that, the authors should demonstrate in Figure 1, what is the EGP, fecundity and parasite burden of a single parental (rat-adapted) strain exposure in STAT-6 KO mice in comparison with the STAT-6 adapted strain?

3) In the same line, G1 from STAT-6 KO when infects WT mice also elicit a stronger immune response?

4) Is there any difference in the parasite burden of a rat-adapted strain infecting the lungs of WT vs STAT-6 KO mice?

5) In figure 6 and 7, the characterization of immune response could be more explored by investigating the mechanisms described in the literature associated with the protective immunity against Nippo. Did the authors measure eosinophils? Mucus production?

**Part III – Minor Issues: Editorial and Data Presentation Modifications**

Reviewer #1: Figure 1D: WT vs STAT6 KO data is in the opposite order to other figures

Figure 2A: STAT6KO worm image contains a section of background medium (outside of the boundary of the worm edge), making it look redder and bigger, and is slightly misleading.

Lines 234-267: The genes mentioned here could be highlighted in volcano plots or shown as raw data. As a resource for the community, this raw data should be made available.

Supp Fig 7A-C: FcERI+ cells could be mast cells or DCs, could presumably tell by looking at SSC to see whether there are the same populations, or increased recruitment of a specific cell type?

There is a placeholder statement for the data availability, presumably this will be corrected later.

Reviewer #2: The authors found that STAT6 KO-conditioned parasites induce higher inflammation when used to infect WT animals, however, discuss little about the mechanisms. They can discuss parasite-related factors, since they have the transcriptomics and know the few genes that are still differentially expressed. In addition, they could provide some information about how these parasites regulate anti-inflammatory mechanisms on the host. They have FOXP3 on the flow cytometry panels, but do not show any information on Tregs, which are relevant to these findings.

Another intriguing finding is that STAT6KO-adapted L4 showed minimal adaptation (G1), when compared to adult worms at the same passage. This is coherent with the finding that despite such adaptation in adults, no difference in survival is observed when adapted worms are reintroduced to WT mice. It would be interesting if the authors could provide some discussion on the different mechanisms at play in the lungs and intestines that are dependent on STAT 6 and why there is such a big difference between L4 and adults, or would it be just because L4 activated less genes during their development and are less likely to modulate large numbers of genes based on host’s immune pressure.

A minor comment is that it is hard to distinguish blue dots from black dots in Figure 3 and alike. Perhaps using a lighter blue would facilitate to appreciate the results easily.

Reviewer #3: (No Response)

PLOS authors have the option to publish the peer review history of their article (what does this mean?). If published, this will include your full peer review and any attached files.

Reviewer #1: No

Reviewer #2: **Yes: **Helton Santiago

Reviewer #3: No
---

## [Decision Letter · Decision Letter 1]

2 Nov 2023

Dear Dr. Herbert,

We are pleased to inform you that your manuscript 'Hookworms dynamically respond to loss of Type 2 immune pressure' has been provisionally accepted for publication in PLOS Pathogens.

Best regards,

Thomas B. Nutman

Academic Editor

PLOS Pathogens

James Collins III

Section Editor

PLOS Pathogens

Kasturi Haldar

Editor-in-Chief

PLOS Pathogens

orcid.org/0000-0001-5065-158X

Michael Malim

Editor-in-Chief

PLOS Pathogens

orcid.org/0000-0002-7699-2064

Reviewer Comments (if any, and for reference):

Reviewer's Responses to Questions

**Part I - Summary**

Reviewer #3: The authors have effectively addressed all the comments and concerns raised during the peer-review process. I highly recommend that the editorial board accept this revised version for publication.

**Part II – Major Issues: Key Experiments Required for Acceptance**

Reviewer #3: (No Response)

**Part III – Minor Issues: Editorial and Data Presentation Modifications**

Reviewer #3: (No Response)

PLOS authors have the option to publish the peer review history of their article (what does this mean?). If published, this will include your full peer review and any attached files.

Reviewer #3: No

---

## [Editor Report · Acceptance letter]

6 Dec 2023

Dear Dr. Herbert,

We are delighted to inform you that your manuscript, "Hookworms dynamically respond to loss of Type 2 immune pressure," has been formally accepted for publication in PLOS Pathogens.

Best regards,

Michael Malim

Editor-in-Chief

PLOS Pathogens

orcid.org/0000-0002-7699-2064